# Do community-based active case-finding interventions have indirect impacts on wider TB case detection and determinants of subsequent TB testing behaviour? A systematic review

Helena R. A. Feasey[1,2]*, Rachael M. Burke[1,2], Marriott Nliwasa[3], Lelia H. Chaisson[4], Jonathan E. Golub[5], Fahd Naufal[5], Adrienne E. Shapiro[6], Maria Ruperez[2], Lily Telisinghe[2,7], Helen Ayles[2,7], Cecily Miller[8], Helen E. D. Burchett[9], Peter MacPherson[1,2,10], Elizabeth L. Corbett[1,2]

1 Malawi-Liverpool-Wellcome Trust Clinical Research Programme, Blantyre, Malawi, 2 TB Centre, London School of Hygiene and Tropical Medicine, London, London, 3 College of Medicine, University of Malawi, Blantyre, Malawi, 4 Division of Infectious Diseases, Department of Medicine, University of Illinois at Chicago, Chicago, IL, United States of America, 5 Center for Tuberculosis Research, Department of Medicine, Johns Hopkins University, Baltimore, MD, United States of America, 6 Department of Global Health and Department of Medicine, University of Washington, Seattle, WA, United States of America, 7 Zambart, University of Zambia School of Public Health, Ridgeway, Zambia, 8 WHO, Geneva, Switzerland, 9 Faculty of Public Health & Policy, London School of Hygiene and Tropical Medicine, London, United Kingdom, 10 Liverpool School of Tropical Medicine, Liverpool, United Kingdom

* helena.feasey@lshtm.ac.uk

## Abstract

Community-based active case-finding (ACF) may have important impacts on routine TB case-detection and subsequent patient-initiated diagnosis pathways, contributing "indirectly" to infectious diseases prevention and care. We investigated the impact of ACF beyond directly diagnosed patients for TB, using routine case-notification rate (CNR) ratios as a measure of indirect effect. We systematically searched for publications 01-Jan-1980 to 13-Apr-2020 reporting on community-based ACF interventions compared to a comparison group, together with review of linked manuscripts reporting knowledge, attitudes, and practices (KAP) outcomes or qualitative data on TB testing behaviour. We calculated CNR ratios of routine case-notifications (i.e. excluding cases identified directly through ACF) and compared proxy behavioural outcomes for both ACF and comparator communities. Full text manuscripts from 988 of 23,883 abstracts were screened for inclusion; 36 were eligible. Of these, 12 reported routine notification rates separately from ACF intervention-attributed rates, and one reported any proxy behavioural outcomes. Two further studies were identified from screening 1121 abstracts for linked KAP/qualitative manuscripts. 8/12 case-notification studies were considered at critical or serious risk of bias. 8/11 non-randomised studies reported bacteriologically-confirmed CNR ratios between 0.47 (95% CI:0.41–0.53) and 0.96 (95% CI:0.94–0.97), with 7/11 reporting all-form CNR ratios between 0.96 (95% CI:0.88–1.05) and 1.09 (95% CI:1.02–1.16). One high-quality randomised-controlled trial reported a ratio of 1.14 (95% CI 0.91–1.43). KAP/qualitative manuscripts provided

**Data Availability Statement:** All data is available within the results and supplementary materials tables.

**Funding:** This work was made possible through grants provided by the WHO Global TB Programme. RMB, ELC, and PM hold Wellcome fellowships: 203905/Z/16/Z (RMB), 200901/Z/16/Z (ELC), and 206575/Z/17/Z (PM). MR, LT, and HA are funded by part of the European and Developing Countries Clinical Trials Partnership 2 programme supported by the EU (grant number RIA2016S-1632-TREATS). AES is supported by a National Institutes of Health (NIH) grant K23AI140918. WHO facilitated discussions among authors at the design stage and contributed to this manuscript but had no role in the conduct or writing of the WHO review. Wellcome, European and Developing Countries Clinical Trials Partnership, and NIH had no role in the design or conduct of this review.

**Competing interests:** I have read the journal's policy and the authors of this manuscript have the following competing interests: JEG, HA, and ELC are authors of trials included in this systematic review. HA and ELC are members of the WHO TB Screening Guideline Development Group, which CM co-ordinates. JEG, HA, ELC, and PM have received research grants to their institutions for projects evaluating community-based active case-finding. All other authors declare no competing interests.

insufficient evidence to establish the impact of ACF on subsequent TB testing behaviour. ACF interventions with routine CNR ratios >1 suggest an indirect effect on wider TB case-detection, potentially due to impact on subsequent TB testing behaviour through follow-up after a negative ACF test or increased TB knowledge. However, data on this type of impact are rarely collected. Evaluation of routine case-notification, testing and proxy behavioural outcomes in intervention and comparator communities should be included as standard methodology in future ACF campaign study designs.

## Introduction

With over 1.4 million deaths per year [1], tuberculosis (TB) was second only to SARS-CoV-2 as an infectious cause of death globally in 2020. As many as three million people are living with undiagnosed TB disease [1]. Early diagnosis and treatment are fundamental to TB control efforts: the WHO End TB strategy includes targets of at least 90% of people who develop TB being notified and treated within one year by 2025 [2]. Innovative approaches are needed to accelerate progress towards this target from the current estimate of 71% [1].

WHO defines both patient-initiated care-seeking and provider-initiated systematic screening approaches to identify people living with undiagnosed TB [3, 4]. Screening pathways can be facility-based systematic screening or community-based "active case-finding" (ACF). Patient-initiated care-seeking can arise through people recognizing TB symptoms and presenting to a health facility (passive case-finding or PCF), or result from advocacy, communication and social mobilization activities (ACSM) that can prompt earlier care seeking for facility-based TB screening (enhanced case-finding or ECF).The key difference between ACF and ECF is that ACF implies individual interaction between a participant and healthcare worker in the community (e.g. where the participant completes a symptom screen, submits sputum for TB testing or undergoes a chest X-ray).

ACF interventions are designed to directly identify people living with undiagnosed TB in the community but may also have an indirect impact on wider TB case detection as seen in an 2011–14 ACF intervention in Blantyre, Malawi where routine facility-based case-notifications increased substantially over the intervention period [5]. Routine case notification rate (CNR) ratios with a comparison group (excluding those directly identified through the ACF) >1 would be an indication of indirect impact. This indirect impact could be due to enhanced diagnostics introduced through the intervention or an impact on subsequent community TB testing rates and behaviour. Enhanced diagnostics could increase routine case-notifications through improved test sensitivity, although this is likely to be limited to bacteriologically-confirmed TB and there may be a concurrent drop in clinically-diagnosed TB. Health workers could also offer more TB tests if aware of the enhanced diagnostic capacity, leading to higher testing rates. Higher TB testing rates could be due to changes in health worker or community behaviour.

ACF interventions can cover a wide range of activities including door-to-door visits or mobile clinics. They are almost invariably accompanied by ACSM activities, even if only to promote ACF participation and explain to the community the purpose of the intervention and the need for repeat testing if symptoms persist. As such, ACF could influence subsequent TB testing behaviour through the three elements of the COM-B behavioural theory (capacity, opportunity and motivation) and potentially increase TB case-notifications in health facilities through indirect effects (Fig 1). COM-B is a comprehensive model developed from a review of 19 existing behaviourial theories [6] that has been widely applied in assessing and developing

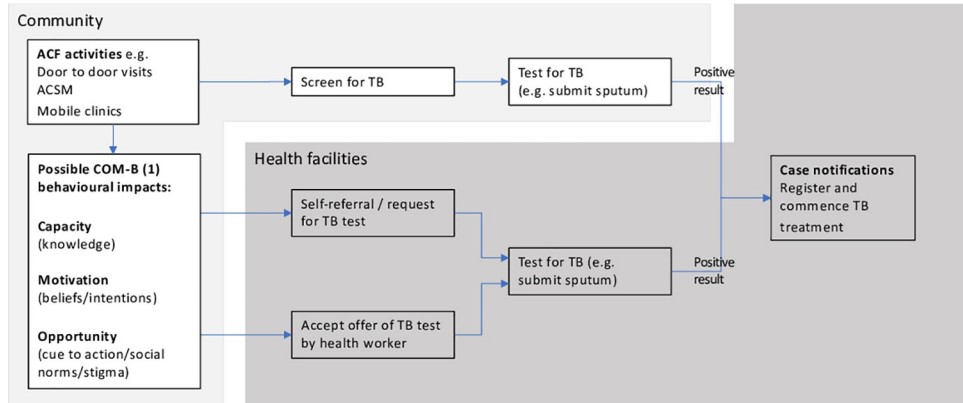

**Fig 1. Conceptual framework for how tuberculosis active case finding may affect subsequent healthcare-seeking behaviour.** Footnote: (1) Capacity, Opportunity and Motivation are the three domains of the COM-B behavioural theory [6].

public health interventions [6–9] including those for Tuberculosis diagnosis and prevention [10–12].

The behavioural mechanisms by which ACSM delivered through ACF interventions may lead to increased knowledge about TB disease and services, or act as a prompt for symptomatic people to present to a health centre for TB testing, are not well understood. ACF interventions could affect knowledge, attitudes and practice (KAP), prompting more timely care-seeking and increasing levels of TB testing and case-notifications through health facilities. ACF interventions may also reduce TB stigma or change social norms and community perceptions around TB. These factors could influence the capacity, motivation and opportunity [6] for subsequent TB testing behaviour (Fig 1). The duration of any behaviour change from ACF is likely to be modified by characteristics of the target population, such as level of education, and ease of access to routine healthcare.

Previous systematic reviews by Kranzer et al (2013) [13], Mhimbira et al (2017) [14] and Burke et al (2021) [15] have shown that ACF interventions can initially increase TB case-notifications, but not invariably. The indirect effects of ACF on routine case-notifications however, has not previously been reviewed. We therefore aimed to systematically review the evidence of indirect effects of ACF on routine facility-based TB case-notifications and also accompanying quantitative proxy behavioural outcomes, such as KAP, that could inform the mechanisms underlying any effect on subsequent TB testing behaviour.

## Methods

We conducted a systematic review of studies reporting the indirect effect of community ACF for TB on routinely-diagnosed TB case-notifications and quantitative proxy behavioural outcomes, such as self-reported TB testing behaviour and KAP of TB.

### Definitions

**Active case finding (ACF)** was defined as systematic TB screening activities implemented in a specific population. The screening could take any form (e.g. symptom interview, radiology, microbiological testing, referral for specialist medical assessment, in any order) but required a personal interaction between a screener and the person being screened. Health promotion communication activities alone (e.g. leaflet delivery) were considered to be ECF and not ACF. Interventions based solely at a routine healthcare facility were considered systematic TB screening interventions, not ACF.

**Routinely-diagnosed TB case-notifications** were those identified through ongoing standard healthcare facility-based case-finding activities and excluding TB case-notifications identified through ACF activities (whether tested in the field or referred for testing after screening in the community).

**Additionality** represents the total increase above expected numbers in TB case-notifications following an active case-finding intervention. This captures all patients who would not have been identified during that time period in the absence of the intervention [16], and can be estimated from comparison of changes in case-notifications in the intervention population during the project compared to the control population or period [17].

**Substitution** represents the phenomenon of TB patients diagnosed by an active case-finding intervention who, in the absence of the intervention, would still have been identified through routine case-finding activities within the same time period. The extent to which substitution has occurred can be estimated from the number of patients directly diagnosed by ACF minus those identified as additional cases (additionality).

The quantitative proxy behavioural outcomes we examined were:

**TB knowledge, attitudes and practices (KAP)** were what is known, believed and done in relation to TB [18], typically assessed through pre- and post-intervention surveys.

**Testing for TB** was when a person who has TB symptoms or signs suggestive of TB has a diagnostic test (through submitting sputum for microbiological testing, radiology or specialist medical assessment).

**TB stigma** was defined as a dynamic process of devaluation that significantly discredits an individual in the eyes of others due to their known or suspected TB status. Within particular cultures or settings, certain attributes are defined by others as discreditable or unworthy [19]. TB stigma could be assessed through a validated scale or through qualitative data.

**TB social norms** were rules and standards that are understood by members of a group, and that guide or constrain social behaviours around TB, without the force of law [20]. Social norms could be assessed through quantitative data using validated domains or vignettes, or qualitative approaches.

## Inclusion and exclusion criteria

We included studies evaluating an ACF intervention that compared epidemiological TB outcomes (TB case-notifications or TB prevalence) between populations exposed and not exposed to ACF and reported either routinely-diagnosed TB case-notifications or identified proxy behavioural outcomes. Routinely-diagnosed TB notification outcomes could either be directly reported or calculated if both direct ACF yield and overall case notifications were reported for the same period and relevant population. Applicable study designs included randomised controlled trials, studies with a parallel comparison group (controlled before-after studies) and studies with a time-based comparison (before-after studies). We included studies with adults aged 15 years or older that screened at least 1000 people (since the prevalence of active TB in a community will rarely exceed 1%). Interventions conducted in closed communities (e.g. prisons) and specific occupational groups (e.g. miners) were included but screening interventions for contacts of people with TB (contact tracing) were not. Studies published before 1 January 1980 and those not in English were excluded.

## Search strategies

The literature search included all studies identified in a previous review by Kranzer et al in 2013 [13], covering the period 1 Jan 1980 to Oct 13 2010, and an additional search of PubMed, EMBASE, Scopus and the Cochrane Library for papers published between 1 Nov 2010 and 4

Feb 2020 (subsequently updated to 13 April 2020) (search strategy in S1 Text). Studies identified through the updated search were title and abstract double screened for initial eligibility (original research, where ACF had taken place, written in English, French or Spanish) by FN, AES and LHC. The full text of eligible studies and all studies from the Kranzer and colleagues review were reviewed by two of HRAF, RMB and MN. Inclusion decisions were resolved by consensus and discussion with ELC and PM. Reference lists from eligible manuscripts were examined and expert opinion on other available papers was sought from members of the WHO TB Screening Guideline Development Group for this and the accompanying review on TB ACF effectiveness [15]. Data was extracted from studies independently by two of HRAF, RMB and MN and entered into a spreadsheet.

## Accompanying qualitative and KAP studies literature search

To increase the number of studies reporting proxy behavioural outcomes relevant to subsequent health seeking behaviour, a further search was conducted for additional secondary manuscripts on qualitative or KAP studies related to the ACF studies identified through the initial literature search (search strategy S2 Text). To be included, the study had to be part of the ACF intervention study identified through the main literature search and include qualitative or quantitative data on the impact of the ACF itself on community TB health seeking behaviour (KAP, TB testing behaviours, pathways to care, TB stigma or social norms). Studies not specifically demonstrating the impact of the ACF on these factors in the ACF target population were excluded, e.g. if the KAP measures were for a different population.

## Access to healthcare

We classified studies according to level of healthcare access within the target population based on distance to and cost of care, as indicated by the reported context or assumed from knowledge of the local health system (S3 Table), on a scale of 'Standard' (routine free healthcare available within catchment area), 'Restricted' (access restricted due to distance and/or cost) or 'Hard to reach' (populations specifically selected as hard to reach).

## Outcomes and risk of bias assessment

Outcomes were a comparison of routine case notification rates (excluding those identified through ACF) and a comparison of reported TB KAP scores (proxy behavioural measure) between groups exposed to and not exposed to the community-based ACF.

To establish routinely diagnosed case notification rates, person-years of follow-up and notified TB cases diagnosed only through routine screening activities were extracted or calculated from available data using simple arithmetic (see S3 Table for extracted data). Person-years were calculated for the target populations for which case-notifications were reported. For before-after studies if the size of the population was not reported separately for the pre- and post-intervention periods it was assumed the size of the population did not change. None of the studies presented case-notification ratios for routine diagnosis; we calculated these through subtracting the available ACF-specific case-notifications from the overall notification data. For randomised and before-after studies we calculated the CNR ratio (intervention vs control groups or baseline vs post-intervention populations) and for controlled before-after studies with a non-randomised comparison group the outcome measure was a comparison of the before to after TB CNR ratio in the two comparison groups: the ratio of the CNR ratios.

Where data was available confidence intervals were calculated using Stata. For studies affected by clustering, three possible values (0.01, 0.05 and 0.1) of the intra-cluster correlation coefficient (ICC) were used to calculate three possible 95% confidence intervals using the

Cochrane recommended method [21]. Only the narrowest intervals (ICC = 0.01) are presented in this text, with the others presented in Table 2. Confidence intervals for KAP scores are presented as reported by the authors.

For randomised studies, the Cochrane Risk of Bias (ROB) tool [22] was used to assess risk of bias. Non-randomised studies were assessed for risk of bias using ROBINS-I [23] and qualitative studies were assessed through the Critical Appraisal Skills Programme (CASP) checklist [24].

### Ethical approval and data availability

Ethical approval was not required for this study. All data is available within the results and supplementary materials tables.

## Results

From a total of 23,883 studies identified, full texts of 988 were assessed for inclusion (S1 Table), and 36 with a suitable community-based ACF study design for this review were identified, including 12 that reported case-notification data from both routine facilities and from ACF-identified notifications (Fig 2). Only one out of the 36 manuscripts reported any proxy behavioural outcomes [25], but the additional search identified 1121 manuscripts, of which four articles were eligible for inclusion as KAP/qualitative manuscripts after full text review, but two of these were excluded from full analysis following identification of additional documentation (S2 Table).

### Routine TB case-notifications

Of the 12 studies identified for the review of ACF impact on routinely-identified case-notifications, one was a randomised controlled trial [26], six were controlled before-and-after studies (with a parallel comparison group) and five were before-after studies with no comparison group (Table 1). One of the controlled before-and-after studies (Cegielski 2013 [27]) was excluded from further analysis since no cases of TB were identified after the intervention period so meaningful case notification ratios could not be calculated. For all studies (except Miller 2010) the "after" or outcome notifications period was the period during the intervention and did not extend beyond.

Populations varied from urban high-density neighbourhoods to rural communities with long distances to healthcare. From the limited information available, three studies were classified as having been conducted in a setting with "standard" access to routine healthcare, two were classified as specifically "hard-to-reach" and the rest were classified as having restricted access to routine care due to remoteness and/or cost (see S3 Table for extracted data).

ACF interventions combined different strategies including door-to-door screening (eight studies), sputum collection by volunteers or community health workers (seven studies) and community mobilisation for mobile screening clinics (four studies) (Table 1). Of the 11 studies analysed, four reported only bacteriologically-confirmed TB, two reported data for all forms TB (including clinically-diagnosed TB) and five reported both. Only Datiko 2017 and Lorent 2014 reported improving routinely available TB diagnostics as part of the intervention.

The included RCT was conducted by Miller et al comparing door-to-door ACF with leaflet delivered ECF in a Brazilian favela, with a staggered intervention delivered serially in pairs of clusters [26]. The total trial period was 283 days, including the complete intervention time through 60 days after ending ACF in the final clusters. Using calendar time-period, the CNR ratio was 1.14 (95% CI: 0.94–1.40) implying a 14% relative increase in non-ACF-diagnosed case notification rate for ACF compared to ECF (Table 2). A before-during-after analysis,

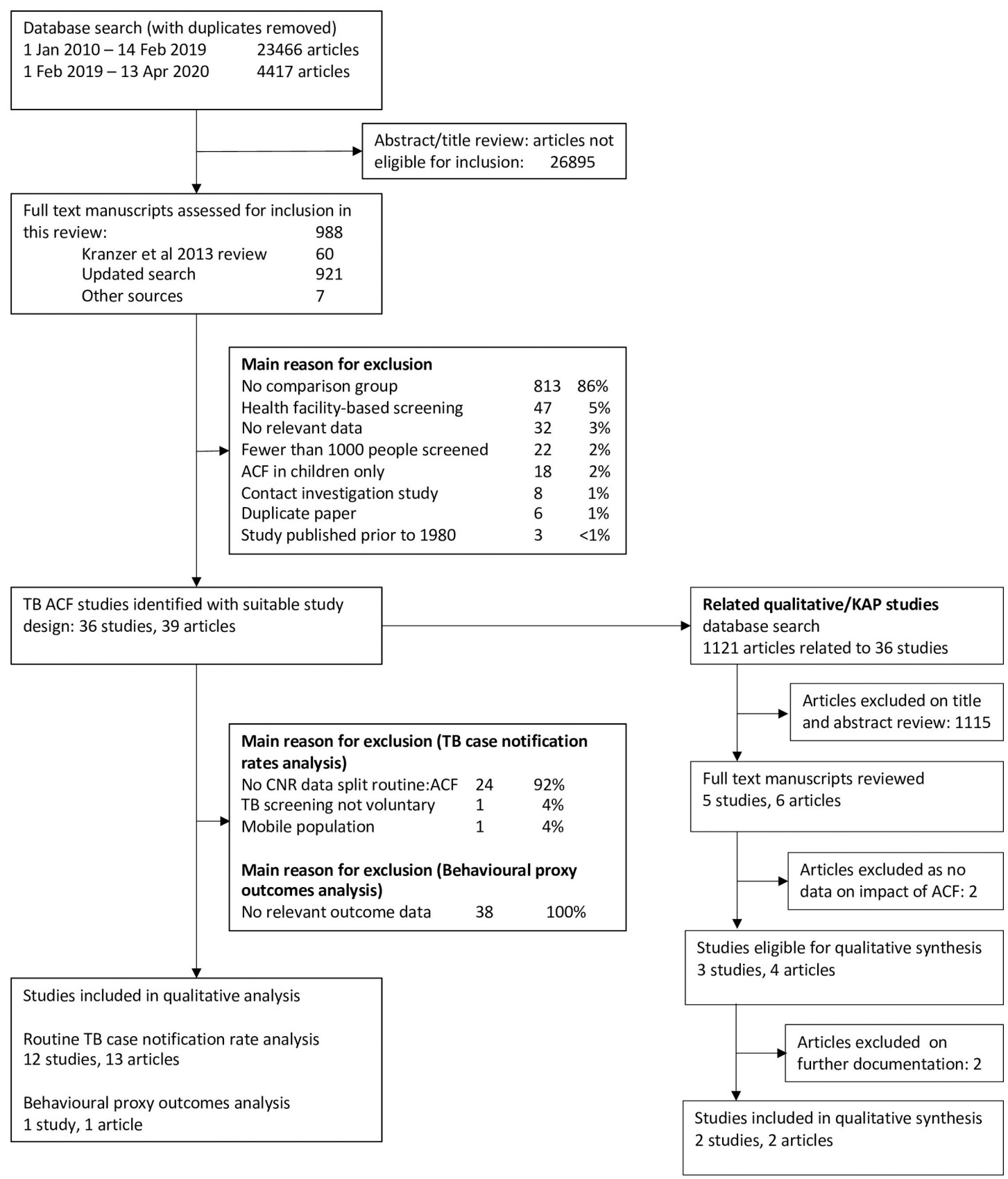

**Fig 2. Modified PRISMA diagram showing articles reviewed and main reasons for exclusion.**

however, accounting for the staggered cluster pair-by-pair initiation design, showed data consistent with a degree of "substitution" whereby patients who would otherwise have been diagnosed routinely during the intervention period and immediately afterwards were found though ACF. The CNR ratio for ACF compared to ECF clusters was, 0.65 (95% CI: 0.36–1.19) during the intervention and 0.80 (95% CI: 0.51–1.27) for the 60 days immediately after the intervention, but 1.42 (95% CI: 1.12–1.82) for days outside this period (both pre intervention and >60days to end of follow-up) which accounted for 68.5% of the 283-day total trial period (Fig 3). There were some concerns of bias due to missing data in this study.

Of the other included studies, the outcome measure of routinely-diagnosed CNR ratio or ratio of CNR ratios (depending on study design) ranged from 0.96 to 1.09 for all forms of TB and 0.47 to 0.96 for bacteriologically-confirmed TB (Table 2). These differences were only significant at the p<0.05 level for three of the seven studies reporting all forms of TB: Aye 2018 1.09 (95% CI: 1.02–1.16) [28], Fatima 2016 1.04 (95% CI: 1.03–1.05) [29] and Fatima 2014 1.06 (95% CI: 1.03–1.09) [30]. Confidence intervals were not calculated for Ford 2019 [31] due to

**Table 1. Characteristics of included studies.**

| Study | Design | Country | Population | Healthcare access | ACF | Qualitative / KAP studies |
|---|---|---|---|---|---|---|
| **Case-notifications outcomes** | | | | | | |
| Miller 2010 | Cluster-randomised trial | Brazil | Urban slums | Standard | ACF (door to door) vs. usual case finding plus leafleting | |
| Aye 2018 | Controlled before-after | Myanmar | Urban slums (& "neighbourhood contacts") | Standard | Door to door symptom screening and sputum collection for "neighbourhood contacts", community mobilisation and sputum collection for others | |
| Cegielski 2013 | Controlled before-after | USA | General population—urban | Standard | Community mobilisation, TST screening, mobile clinic. | |
| Datiko 2017 (& Yassin 2013) | Controlled before-after | Ethiopia | Remote rural | Restricted | Community mobilisation, door to door symptom screening, sputum transport | Tulloch 2015 |
| Kan 2012 | Controlled before-after | China | General population—rural | Restricted | Schoolchildren reported symptoms in family members, home visits to symptomatic people, sputum transport. | |
| Parija 2014 | Controlled before-after | India | General population—rural | Restricted | Community mobilisation, mobile clinic, community health workers | |
| Vyas 2019 | Controlled before-after | India | Indigenous groups | Restricted | Door to door symptom screening, sputum collection | |
| Corbett 2010 | Before-after | Zimbabwe | General population—urban | Standard | Community mobilisation, door to door symptom screening or mobile clinics | |
| Fatima 2016 | Before-after | Pakistan | Urban slums "neighbourhood contacts" | Standard | Door to door, sputum collection. | |
| Fatima 2014 | Before-after | Pakistan | Urban slums perceived high risk or hard to reach | Hard to reach | Community mobilisation, mobile clinics (microscopy) | |
| Ford 2019 | Before-after | India | Remote rural | Restricted | Community mobilisation, mobile clinics (CxR). | |
| Lorent 2014 | Before-after | Cambodia | Urban slums—perceived high risk or hard to reach | Hard to reach | Community health workers, door to door symptom screening, sputum collection | Lorent 2015 |
| **Behavioural outcomes (KAP)** | | | | | | |
| Adane 2019 | RCT | Ethiopia | Prison | N/A | Peer educators in prisons. People in prison with identified TB symptoms in control and intervention transferred to clinic for physician review | |

**Table 2. Routinely-diagnosed TB case-notifications outcome measures.**

| Study | Healthcare access | CNR Ratio / Ratio of CNR ratios* | 95% CI† ICC = 0.01 | 95% CI ICC = 0.05 | 95%CI ICC = 0.10 |
|---|---|---|---|---|---|
| **Randomised controlled trial (RCT)** | | | | | |
| Miller 2010 | Standard | 1.14 | 0.94–1.40 | 0.72–1.76 | 0.58–2.15 |
| **Controlled before-after trials–bacteriologically confirmed** | | | | | |
| Datiko 2017 | Restricted | 0.47 | 0.41–0.53 | - | - |
| Kan 2012 | Restricted | 0.81 | 0.66–0.99 | 0.52–1.32 | 0.42–1.61 |
| Parija 2014 | Restricted | 0.85 | 0.77–0.94 | 0.67–1.06 | 0.60–1.15 |
| Vyas 2019 | Restricted | 0.83 | 0.77–0.88 | 0.71–0.97 | 0.66–1.04 |
| **Controlled before-after trials–all forms** | | | | | |
| Aye 2018 | Standard | 1.09 | 1.02–1.16 | 0.94–1.27 | 0.88–1.35 |
| Datiko 2017 | Restricted | 0.96 | 0.88–1.05 | - | - |
| Vyas 2019 | Restricted | 1.00 | 0.95–1.05 | 0.90–1.12 | 0.86–1.18 |
| **Before-after trials–bacteriologically confirmed** | | | | | |
| Corbett 2010 | Standard | 0.75 | 0.63–0.89 | - | - |
| Fatima 2016 | Standard | 0.96 | 0.94–0.97 | - | - |
| Fatima 2014 | Hard to reach | 0.93 | 0.90–0.95 | - | - |
| Lorent 2014 | Hard to reach | 0.83 | 0.77–0.89 | - | - |
| **Before-after trials–all forms** | | | | | |
| Fatima 2016 | Standard | 1.04 | 1.03–1.05 | - | - |
| Fatima 2014 | Hard to reach | 1.06 | 1.03–1.09 | - | - |
| Ford 2019 | Restricted | 1.02 | - | - | - |
| Lorent 2014 | Hard to reach | 0.93 | 0.89–0.97 | - | - |

†For studies not affected by clustering overall confidence interval presented.

ICC values are estimates not from primary study.

unavailability of data. Outcome measures did not appear to be associated with reported healthcare accessibility.

For all five before-after studies, the during intervention overall (both ACF and routine) case notification rates increased but the routine CNR change for bacteriologically-confirmed TB ranged from a 25% reduction (Corbett 2010 [32]) to a 4% reduction (Fatima 2016) (Fig 4), consistent with a degree of substitution or accelerated diagnosis of patients who would otherwise been diagnosed routinely. For all forms of TB, however, the change ranged from a 7% reduction (Lorent 2014 [33]) to a 6% increase (Fatima 2016 [29]). Lorent 2014 was the only before-after study reporting a decrease in all form routine TB CNR during intervention implementation.

For the six controlled before-after studies increases or decreases in the routine TB CNR in the intervention group reflected the directional change in routine case notification rate in the control group for all studies except two bacteriologically-confirmed reports: Parija 2014 [34] (1% increase in control group and 14% reduction in intervention group) and Datiko 2017 [35] (8% increase in control group and 49% reduction in intervention group) (Fig 5). Both studies were conducted with remote rural communities and in Datiko 2017 participants with smear-negative ACF results were offered follow-up radiological TB diagnosis.

The majority of non-randomised studies were considered to be at critical (two studies) or serious risk of bias (6 studies) with three studies at moderate risk of bias (Corbett 2010 [32], Parija 2014 [34] and Vyas 2019 [36]) (Fig 6).

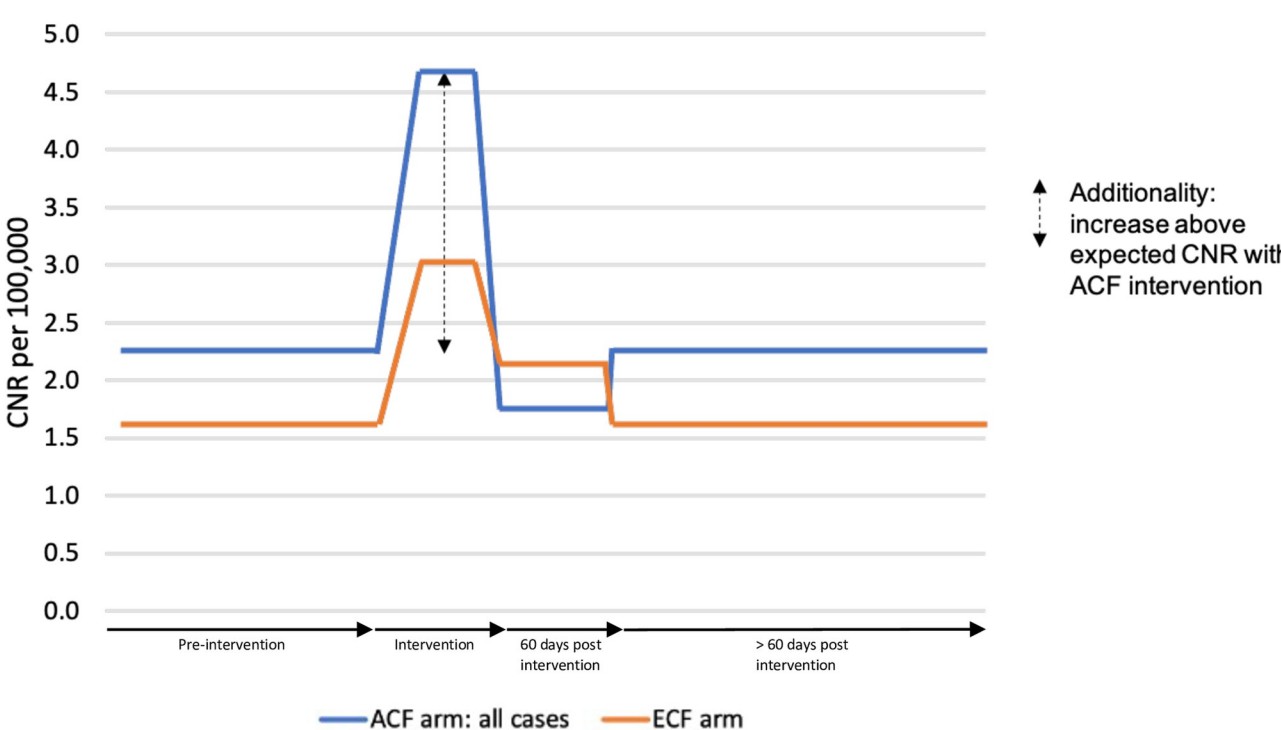

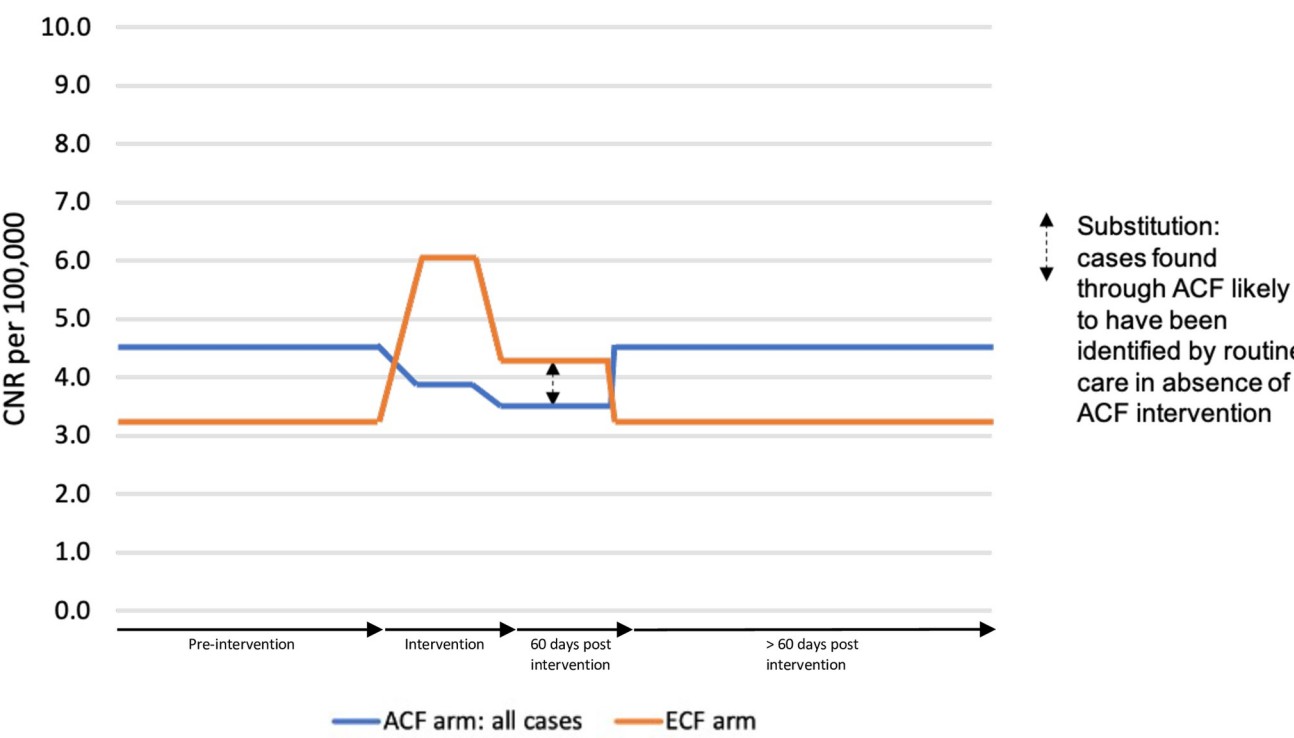

**Fig 3. Case notification rates from Miller cluster-randomised trial in Brazil.** Notes: ACF = Active case-finding; ECF = Enhanced case-finding. Relative CNR in days before intervention and >60 days after intervention unknown so presented as consistent.

### Proxy behavioural outcomes

The included study from the search on proxy behavioural outcomes was a cluster-randomised trial of ACF provided through peer inmate educators in 16 selected prisons in Ethiopia that was classified to be at low risk of bias [25] (Fig 6). KAP scores were collected through a semi-structured post-intervention survey conducted with a randomly selected (process not reported) sample of 1218 inmates, using a pre-tested questionnaire detailed in a separate man-uscript describing questionnaire development and baseline KAP survey results [37].

This study reported that the intervention group had higher levels of good TB knowledge and practice than the control group. Composite scores of overall knowledge (p<0.0001) and good practice (p<0.0001) were significantly higher for ACF compared to control prison respondents, even after adjustment for education, geographical location and cluster size in a generalised estimating equation (GEE) model (adjusted OR 2·54, 95% CI 1·93–3·94 for good knowledge, and adjusted OR 1·84, 1·17–2·96 for good practice). There was no significant

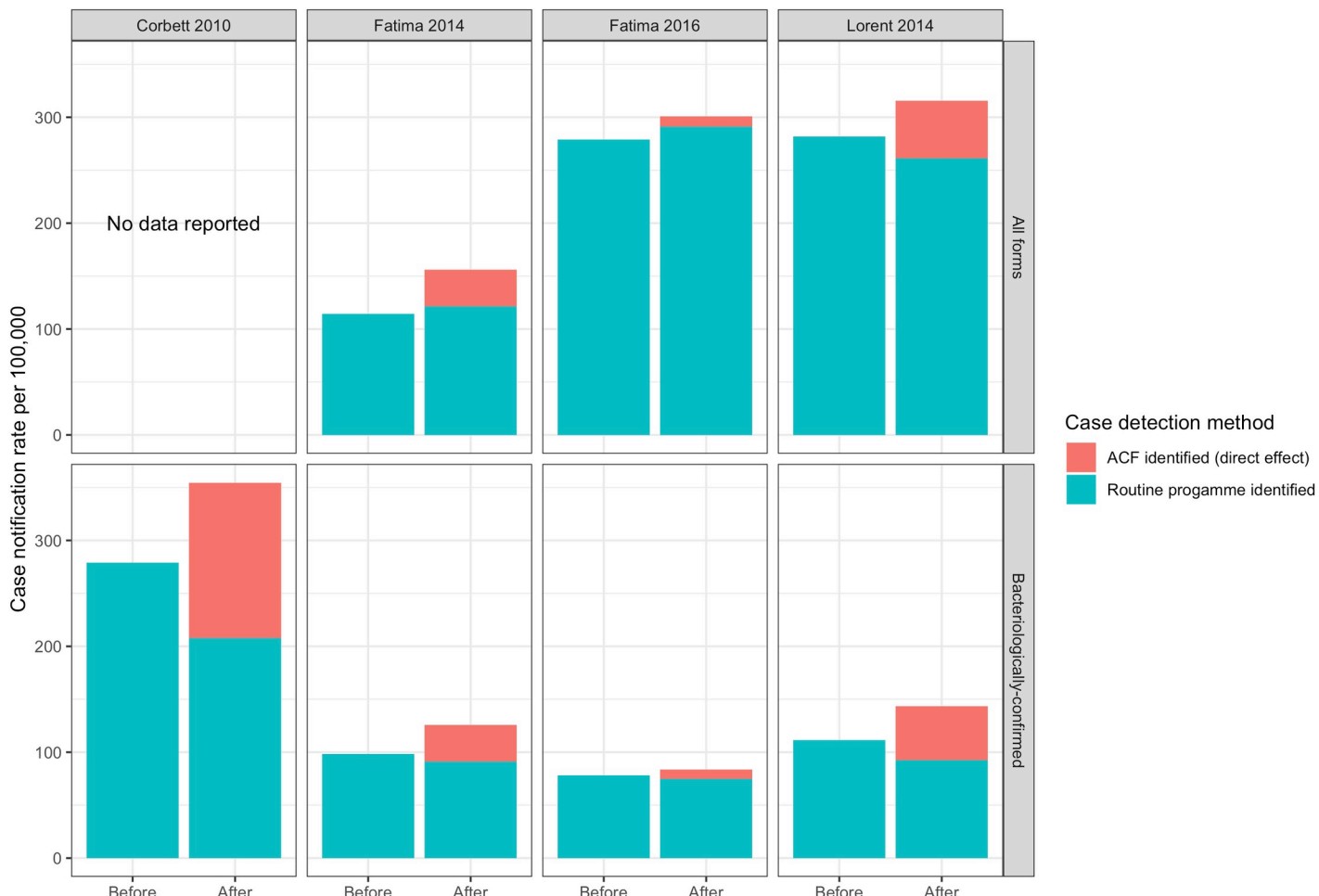

**Fig 4. Routinely diagnosed case notification rates in non-randomised before-after studies.**

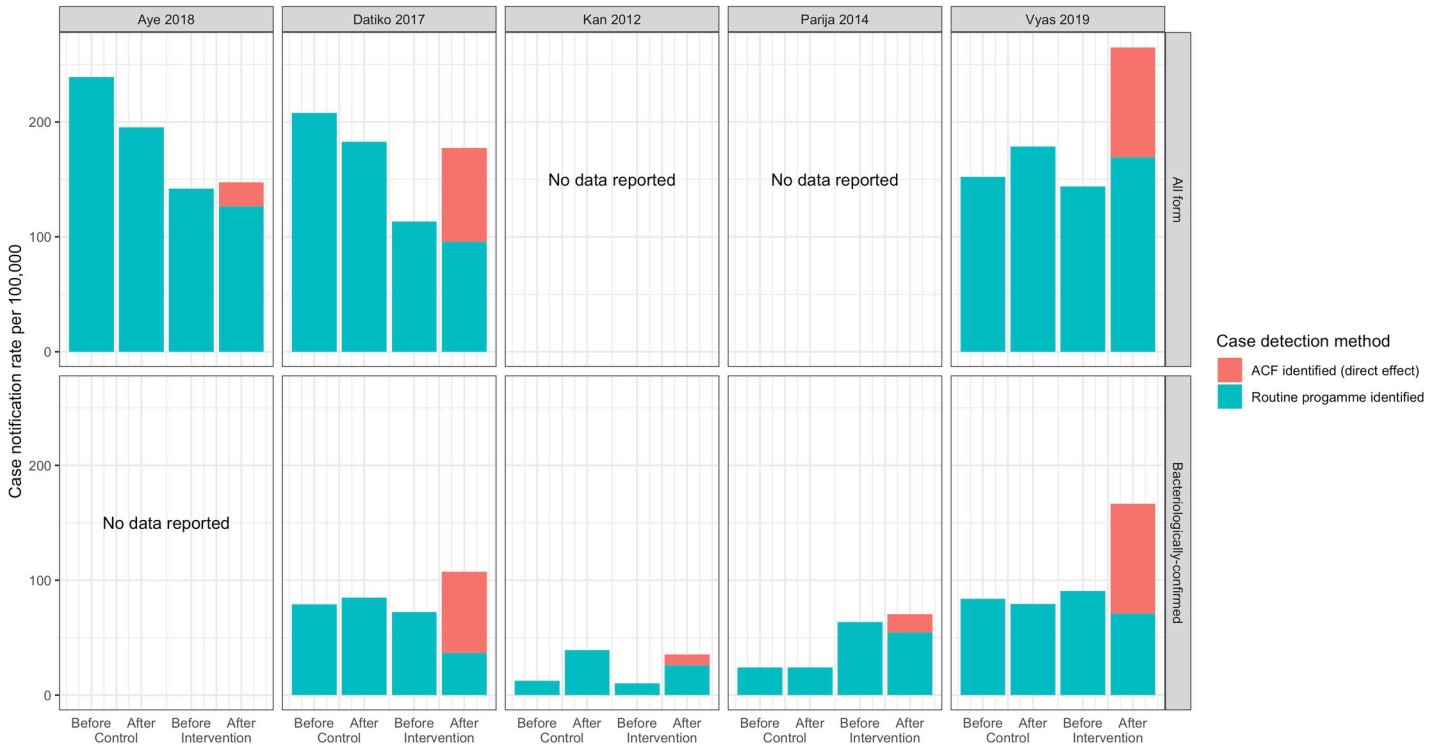

**Fig 5. Routinely-diagnosed case notification rates in controlled before-after studies.**

difference in the composite favourable attitude domain between the two groups (adjusted OR 0.80, 95% CI 0.52–1.25).

## Linked KAP and qualitative studies

Of the four publications [38–41] initially identified, two were excluded from further analysis [38, 39] as additional documentation [42] demonstrated that KAP surveys were not aligned to the populations or timing of the ACF interventions. The two included qualitative studies provided insight into how ACF impacts subsequent TB testing and healthcare-seeking behaviours, although neither directly compared healthcare-seeking behaviours between ACF and routine diagnosis populations.

Tulloch *et al* conducted in-depth-interviews from May 2011 to February 2012 with participants in a door-to-door symptom screening ACF intervention in 19 districts of Sidama zone conducted in rural Ethiopia from Oct 2010 to 2015 [40, 43, 44]. From these data, researchers describe different healthcare-seeking pathways including those who have heard about TB services through the intervention activities, and then self-referred to a facility for testing. Some participants also acted as ongoing advocates: *"There are some who have not heard, if so I always tell them at any opportunity"* [40]. The study thus defines mechanisms through which an indirect effect of the ACF intervention could affect subsequent healthcare-seeking behaviour. In addition, the majority of undiagnosed participants were disappointed to have a negative result with an unresolved health problem: *"I feel much sorrow. I gave them my sputum and they said I was negative but still I feel pain inside... I am not happy about the result."* [40].

Lorent et al. 2015 conducted a survey and interviews with patients diagnosed with TB through door-to-door ACF among high-risk urban populations in Cambodia [33, 41]. Approximately 20% of TB patients diagnosed through the ACF intervention delayed treatment

### a) Randomised trials: Cochrane ROB tool assessment

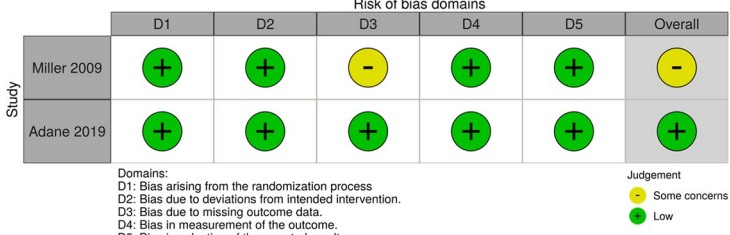

Domains:
D1: Bias arising from the randomization process
D2: Bias due to deviations from intended intervention.
D3: Bias due to missing outcome data.
D4: Bias in measurement of the outcome.
D5: Bias in selection of the reported result.

Judgement
- Some concerns
+ Low

### b) Non-randomised trials: ROBINS-I assessment

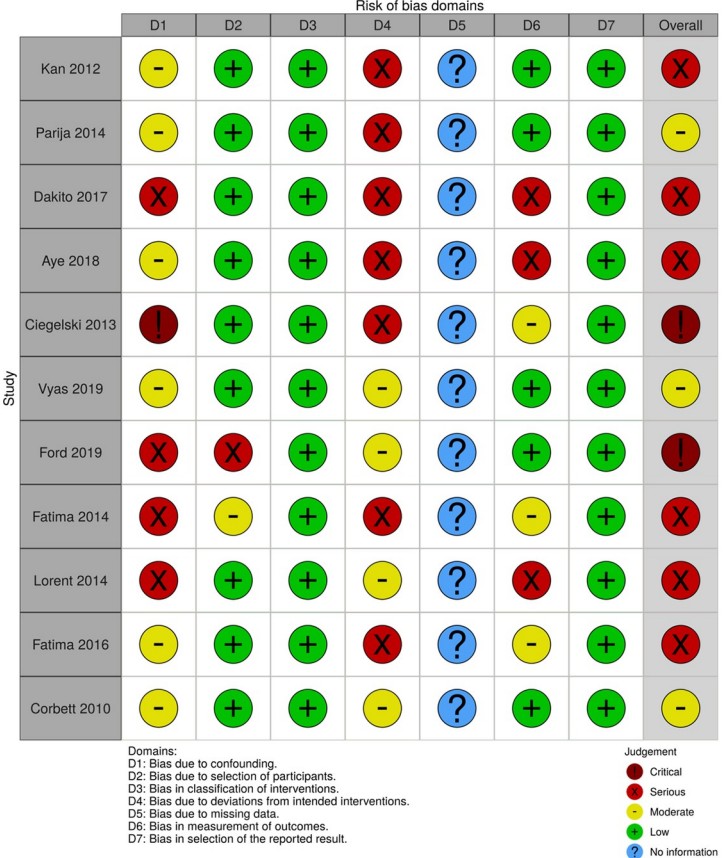

Domains:
D1: Bias due to confounding.
D2: Bias due to selection of participants.
D3: Bias in classification of interventions.
D4: Bias due to deviations from intended interventions.
D5: Bias due to missing data.
D6: Bias in measurement of outcomes.
D7: Bias in selection of the reported result.

Judgement
- Critical
X Serious
- Moderate
+ Low
? No information

### c) Qualitative studies: CASP checklist assessment

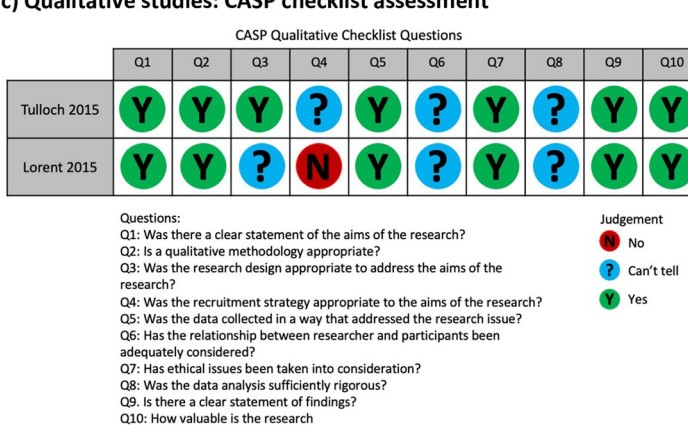

Questions:
Q1: Was there a clear statement of the aims of the research?
Q2: Is a qualitative methodology appropriate?
Q3: Was the research design appropriate to address the aims of the research?
Q4: Was the recruitment strategy appropriate to the aims of the research?
Q5: Was the data collected in a way that addressed the research issue?
Q6: Has the relationship between researcher and participants been adequately considered?
Q7: Has ethical issues been taken into consideration?
Q8: Was the data analysis sufficiently rigorous?
Q9: Is there a clear statement of findings?
Q10: How valuable is the research

Judgement
N No
? Can't tell
Y Yes

**Fig 6. Risk of bias and quality assessments for included studies.**

initiation so the main study focus was on exploring reasons for delayed or failed linkage to care, with a comparison of perspectives between those who delayed treatment initiation and those who started treatment without delay. Participants reported that ACF had removed barriers of access and cost and emphasised the need for health education on TB, including stronger peer-support networks.

## Discussion

To our knowledge, the potential indirect impact of TB active case finding interventions on routine TB case-notifications and subsequent TB testing behaviour has not previously been reviewed. In this systematic review, which has direct relevance to ACF campaigns for other respiratory pathogens such as SARS-CoV-2, we aimed to synthesise evidence from evaluations of TB ACF interventions relating to this indirect, but potentially important, impact. Our main finding was the need for more evidence: we found mixed weak evidence that TB ACF may be effective at indirectly increasing routine TB case notification rates for non-bacteriologically confirmed TB, and insufficient evidence to conclude whether or not ACF impacts subsequent TB testing behaviour. The small number of published studies that specifically address this important issue were at risk of bias introduced by the design or completeness of evaluation, and critical differences in study design precluded meta-analysis as well as firm conclusions. Reaching consensus on how to approach and address this question, including published draft protocols, questionnaires, analysis plans, and key-word suggestions would facilitate the rapid accumulation of high-quality harmonised publications able to support meta-analysis in subsequent systematic reviews. ACF implementers should aim to routinely include prospective qualitative and quantitative assessment of indirect effects, given the critical importance of behavioural change as a key driver of respiratory disease care and prevention [45].

In this review a routine CNR ratio >1 gives an indication of an indirect effect of ACF on routine case-notifications. This was seen in the Miller 2010 RCT (1.14, CI:0.94–1.40) and four of the other studies for all form TB notifications: Aye 2018 (1.09, CI:1.02–1.16), Fatima 2016 (1.04, CI:1.03–1.05), Fatima 2014 (1.06, CI:1.03–1.09), and Ford 2019 (1.02, no CI) but not in any of the bacteriologically-confirmed TB reports. This suggests any indirect impact was unlikely to be due to improved diagnostics implemented through the ACF since this would be expected to be seen primarily in bacteriologically-confirmed rates, but instead may be due to increased TB testing rates and changes in TB testing behaviour. In addition, an indirect effect was not observed in the only two studies which did report improved diagnostics (Datiko 2017 & Lorent 2014). case-notifications. The limited evidence available suggests that there may be a difference in impact between the two forms of TB (Table 2, Figs 4 and 5).

Routine bacteriologically-confirmed TB notifications mostly decreased during the ACF, consistent with a degree of "substitution" (see Methods) whereby ACF identifies some patients who would otherwise have been identified by routine services–although they may have benefited through earlier diagnosis and treatment. Consequently, overall bacteriologically-confirmed CNR increased with ACF but the CNR for routinely diagnosed bacteriologically-confirmed cases decreased (CNR ratio range 0.47–0.96). However, for all forms of TB, routine TB CNRs tended to remain at a similar or slightly higher-level during the community ACF interventions (CNR ratio range 0.93–1.09), which could be explained either by ACF promoting early presentation for clinical diagnosis (when patients are not readily confirmed) or by false positive diagnoses, or a combination of the two.

This difference between bacteriologically-confirmed and all forms TB could be due to the desire identified in Tulloch et al. [40] for participants with negative bacteriological TB results from the ACF to have some resolution for their health problem. These participants could

subsequently attend a facility looking for a diagnosis and then be clinically diagnosed with either extra-pulmonary or pulmonary TB. Datiko et al. [35] and Lorent et al. [33] showed a decrease in routine all forms TB CNR but in the Datiko study, researchers actively followed up ACF participants with negative results by offering them further radiological examination and clinical diagnosis, whilst participants in the Lorent study were selected as the 'most hard-to-reach', suggesting they may have found it difficult to visit a facility for a later clinical diagnosis.

It should be noted that a CNR ratio of ≤1 in this review does not preclude an indirect impact of the ACF on case-notifications as this could still occur but be masked by the "substitution" effect, especially when the CNR ratio is 1 or only slightly below (as in Vyas 2019 (1.00, CI:0.95–1.05), Datiko 2017 (0.96, CI:0.88–1.05) and Lorent 2014 (0.93, CI:0.89–0.97) for all forms TB, and Fatima 2016 (0.96, CI:0.94–0.97) and Fatima 2014 (0.93, CI:0.90–0.95) for bacteriologically-confirmed TB). When the CNR ratio is substantially smaller (e.g. Datiko 2017 (0.47, CI:0.41–0.53) and Corbett 2010 (0.75, CI:0.63–0.89) for bacteriologically-confirmed TB) this suggests there is no indirect impact.

Where it occurs, the indirect impact of ACF on routine TB case-notifications could extend beyond the period of the ACF intervention itself. However, the Miller et al RCT [26] was the only study to specifically assess impact after the end of ACF in a study that reported bacteriologically-confirmed cases only and compared ACF with an ECF intervention. As expected, during the intervention period (mean 27 days) and the 60 days directly afterwards, ECF (leaflets) was associated with increased numbers of TB patients diagnosed through the routine health services. However, the ACF arm had total routine case-notifications beyond those seen with ECF. This could reflect a longer-lasting indirect ACF impact or could just reflect ongoing higher CNRs in the ACF arm since the relative contributions of the pre-intervention and >60 days post-intervention periods are unknown. Personal interaction has been shown to be more effective than purely written information in multiple disciplines [46–48] so temporary in-person community TB diagnosis services could potentially create a longer-term impression than providing literature alone.

We found no evidence that the nature of target populations and levels of healthcare access were important effect modifiers, but cannot conclude that these do not influence the indirect effectiveness of ACF due to the limited number of studies, lack of consistent reporting, and heterogeneity of both populations and interventions.

Disappointingly, we found no studies reporting TB testing rates which would have allowed us to distinguish whether increases in routine TB case-notifications were likely due to an increase in testing or enhanced sensitivity of improved diagnostics with a constant testing rate. In addition, only one study included proxy behavioural outcomes as an integral part of the study design. This Ethiopian cluster randomised trial set in prisons used KAP outcomes as a proxy for subsequent healthcare-seeking behaviour [25] and was assessed as being at low risk-of-bias. TB knowledge and intended care seeking for TB symptoms was improved among inmates provided with the peer-educator intervention, and the study protocol and outcome measures provide a template for subsequent similar interventions and evaluations. Two additional reports provided some qualitative insights supportive of possible impact of ACF on subsequent health seeking behaviour, but conclusions were limited by lack of non-ACF or before-after comparators.

There were several limitations to this review. Despite a literature search covering 40 years and >25,000 titles and abstracts, we found only 12 studies with suitable routine TB case notification data, all of which had very heterogenous interventions and study designs. Just one study specifically addressed outcomes related to subsequent TB testing behaviour following an ACF intervention. As such, we could not conduct meta-analysis, assess generalisability, or quantify the likely impact of behaviour change from ACF on key variables that define the reproduction

number for TB and drive epidemiology [49]. Due to resource and time constraints, we only included manuscripts published in English, and did not include unpublished data or grey literature. Notably, TB REACH (http://www.stoptb.org/global/awards/tbreach/) has funded numerous ACF projects since 2010 with reporting that meets many of our criteria, but we were unable to access unpublished data within the short time available for this review. In addition, the Kranzer et al review used for articles published between 1980 to 2010 did not focus on proxy behavioural outcomes so studies reporting on these could have been missed for this period, but as these outcomes are likely to always be secondary to core outcomes of TB notifications and epidemiology (which were included) the likelihood is low. Statistical limitations include limited availability to adjust for confounders as these data were not consistently reported. We also assumed that ACF diagnoses are a subset of the total notifications but an ACF diagnosis could then become a notification in another population for example through population movement, although this is not reported by any of the studies.

Our main recommendations are to strengthen the evidence regarding ACF and indirect effects on subsequent TB notifications and testing behaviour. Qualitative and quantitative assessment of the indirect effects of ACF should be conducted prospectively. Testing rates would be a better outcome measure than case-notifications to establish indirect impact on TB testing behaviour but these are not routinely collected. Case-notifications, and TB testing where available, from both ACF and routine diagnostic services should be reported separately, ideally including pre-ACF, during-ACF and post-ACF periods, evaluated against a comparator population. The inclusion of a comparator is critical, as this is what allows attribution of impact to the ACF intervention itself. To better understand the mechanisms through which ACF potentially impacts TB testing behaviour, relevant outcomes including TB KAP, test initiator (patient or health worker), stigma and norms should be investigated and reported, ideally through repeated cross-sectional sampling before and after implementation. Accompanying qualitative research would provide the rich detail needed to understand how the ACF intervention creates these indirect impacts on subsequent TB case detection.

## Conclusions

In conclusion, the available literature is insufficient, providing only weak evidence for an indirect effect of ACF on clinically diagnosed routine TB case-notifications and insufficient quantitative evidence to assess whether or not ACF impacts subsequent TB testing behaviour. The few available data suggest that ACF can increase TB knowledge and intention to seek early TB diagnosis, together with a desire for diagnosis in those with negative bacteriological ACF results, with potential to impact on future TB testing and case detection rates. Future ACF intervention studies should incorporate assessment of any indirect impact of ACF on facility-based testing and notifications, and other factors with potential to influence TB testing behaviour including KAP, stigma and social norms.

## Supporting information

**S1 Checklist. PRISMA checklist for systematic review.**
(PDF)

**S1 Text. Main search strategy.**
(PDF)

**S2 Text. Accompanying qualitative and KAP studies search strategy.**
(PDF)

**S1 Table. List of papers about TB ACF reviewed at full text.**
(PDF)

**S2 Table. List of TB ACF studies identified with suitable study design and included in search for additional KAP or qualitative manuscripts.**
(PDF)

**S3 Table. Data extracted from and characteristics of included studies with routine case-notification outcomes.**
(PDF)

## Acknowledgments

We acknowledge Lori Rossman, Pamela Delgado-Barroso, and Hector Alvarez-Manzo (Johns Hopkins University, Baltimore, MD, USA) for their assistance with the database search. We acknowledge the WHO TB Screening Guideline Steering Committee for facilitating discussions among authors at the design stage of this research.

## Author Contributions

**Conceptualization:** Helena R. A. Feasey, Cecily Miller, Peter MacPherson, Elizabeth L. Corbett.

**Data curation:** Helena R. A. Feasey, Rachael M. Burke, Marriott Nliwasa, Lelia H. Chaisson, Fahd Naufal, Adrienne E. Shapiro.

**Formal analysis:** Helena R. A. Feasey, Rachael M. Burke, Marriott Nliwasa, Peter MacPherson, Elizabeth L. Corbett.

**Methodology:** Helena R. A. Feasey, Rachael M. Burke, Peter MacPherson, Elizabeth L. Corbett.

**Project administration:** Helena R. A. Feasey.

**Writing – original draft:** Helena R. A. Feasey, Peter MacPherson, Elizabeth L. Corbett.

**Writing – review & editing:** Helena R. A. Feasey, Rachael M. Burke, Marriott Nliwasa, Lelia H. Chaisson, Jonathan E. Golub, Fahd Naufal, Adrienne E. Shapiro, Maria Ruperez, Lily Telisinghe, Helen Ayles, Cecily Miller, Helen E. D. Burchett, Peter MacPherson, Elizabeth L. Corbett.

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
