## [Decision Letter · Decision Letter 0]

10 Aug 2021

 PGPH-D-21-00221 Do community-based active case-finding interventions affect subsequent tuberculosis testing and healthcare-seeking behaviour? A systematic review PLOS Global Public Health

Dear Dr. Feasey,

Thank you for submitting your manuscript to PLOS Global Public Health. After careful consideration, we feel that it has merit but does not fully meet PLOS Global Public Health’s publication criteria as it currently stands. Therefore, we invite you to submit a revised version of the manuscript that addresses the points raised during the review process.

We look forward to receiving your revised manuscript.

Kind regards,

Stefan Kohler

Academic Editor

Journal Requirements:

Additional Editor Comments (if provided):

Complementary to the reviewers' comments, I would be interested to better understand how the submitted review and its search strategy relate the following recently published review, which seems to use the same search strategy: https://doi.org/10.1016/s2468-2667(21)00033-5. If the same or a similar search strategy was used, why could the same/a similar search strategy be used to systematically review literature on different research questions? What might be strengths and limitations of the chosen approach?

Reviewers' comments:

Reviewer's Responses to Questions

**Comments to the Author**

1. Does this manuscript meet PLOS Global Public Health’s publication criteria? Is the manuscript technically sound, and do the data support the conclusions? The manuscript must describe methodologically and ethically rigorous research with conclusions that are appropriately drawn based on the data presented.

Reviewer #1: Partly

Reviewer #2: Partly

2. Has the statistical analysis been performed appropriately and rigorously?

Reviewer #1: N/A

Reviewer #2: Yes

3. Have the authors made all data underlying the findings in their manuscript fully available (please refer to the Data Availability Statement at the start of the manuscript PDF file)?

Reviewer #1: Yes

Reviewer #2: No

4. Is the manuscript presented in an intelligible fashion and written in standard English?

Reviewer #1: Yes

Reviewer #2: Yes

5. Review Comments to the Author

Reviewer #1: Thank you for having given me the opportunity to review this manuscript. I enjoyed the read!

My main comments concern the search strategy and the conceptual framework that have been used:

- The main search strategy has been copied and updated from Kranzer and colleague's review from 2013. However, the Kranzer review aimed at different research questions (related to number of cases detected, early detection, treatment outcomes, incidence and prevalence), while this review focuses on testing and healthcare seeking. I wonder if the title and research questions of this review may have to be revised to motivate the use of the given search strategy. Moreover, the research questions should be clearly stated.

- The authors mention the COM-B framework and include it in Fig 1. Yet, the motivation for using this framework remains unclear, as well as how exactly it has been applied. I suggest the authors include more information about this in the text and revise Fig 1 for clarity, e.g., clearly highlighting the original elements of the COM-B framework.

Reviewer #2: I think this is an interesting systematic review, and it has been rigorously conducted. However, I had to read it a few times to figure out exactly what was being reviewed in a conceptual sense. I think that the framing of the research question in the introduction and methods needs some revision to make this clear. I also think that some context needs to be provided in multiple places about what the main outcome of routine CNR ratios actually means - that is, what it is potentially telling us, and how a CNR ratio > 1 or <1 should be interpreted.

Major comments

1) The frame of measuring behavioral outcomes of ACF does not seem to fit the systematic review that was performed. In my opinion, not all of the outcomes measured reflect behavior, and the search strategy is not comprehensive for studies of behavior. Regarding the first point, changes in routine case notifications, the primary outcome considered, might not reflect changes in behavior of the population being screened (see comment 2 for more detail). Moreover, it is hard for me to see additionality and substitution as mainly behavioral outcomes since they have as much to do with the presence or absence of the intervention and how it is implemented as the populations’ knowledge, attitudes, or behavior. Regarding the second point, the inclusion criteria of the initial search automatically narrow the studies to those with study designs for quantitative evaluation. This automatically excludes qualitative research, which is often best suited to understanding behavior; qualitative research about the impact of an intervention on participants' behavior would not be done as a pre-post, parallel group comparison, or RCT design. The search for qualitative studies was only done around studies that reported quantitative outcomes, so the search overall is not comprehensive for studies of behavior. I am not suggesting that the authors re-do their systematic review. Rather, I suggest they re-consider the articulation of their frame in the introduction and methods. To me, the systematic review covers quantitative measures of the indirect impact of ACF that could potentially impact case detection separately from the primary ACF intervention, including (but not limited to) the effects on behavior. The discussion actually focuses more on the concept of “indirect effects” as opposed to “behavioral outcomes,” so the issue is really the framing in the introduction and methods.

2) I think that the distinction between overall case notifications (direct impact of ACF) and impact on routine case notifications (indirect impact) needs to be made clearer either in the introduction or earlier in the methods, and the rationale should be given for considering changes in routine case notifications to be a proxy for behavioral outcomes. It is not clear why routine case notifications excluding the direct yield of the ACF intervention are a proxy behavioral outcome. Unlike KAP instruments, which are meant to measure behavior, changes in routine case notifications may not in fact reflect changes in behavior in the population being screened for TB. For example, a positive impact could reflect health system strengthening as a result of an ACF intervention. For example, an ACF intervention may rely on Xpert testing provided in the existing laboratory infrastructure, and to accommodate the increase testing burden, the procurement system for cartridges is strengthened, and procedures for sputum storage and processing are improved. This improvement in the Xpert infrastructure could conceivably end up increasing routine case notifications if it leads to an increase in Xpert testing (or a shift from smear microscopy to Xpert), even if it is not actually changing the behavior of people who are seeking care for TB symptoms. Alternatively, if the framing is revised to shift away from a focus on behavioral outcomes, the introduction/methods still needs some rationale for why changes in routine case notifications excluding direct ACF diagnoses are an outcome of interest. The bulk of the results focus on these CNRs, so it is very important for the reader to understand why they are important.

3) Since CNR ratios are the main outcome, the methods should explicitly describe how they were extracted and the assumptions made in the calculations. The current description “To establish routinely diagnosed case notification rates, person-years of follow-up and notified TB cases diagnosed only through routine screening activities were extracted or calculated from available data using simple arithmetic” is vague. What is the denominator for the case notification rate? The target population or the number of people screened? If target population, and if the size of the target population is not reported separately for the pre-intervention and intervention periods, is the assumption made that the size of the underlying population size does not change (so the ratio of CNR is equivalent to the ration of notifications)? Regarding the numerator for routine CNR, I assume that the authors took the reported total case notifications and subtracted the diagnoses made directly by the ACF intervention. If not, then the calculation needs to be described. If so, then it seems that the assumption is being made that ACF diagnoses are a subset of the total notifications. This assumption is often made in calculation of additionality although it is not completely true, as ACF interventions often diagnose people from outside the target population (e.g. a person visiting from another district may be diagnosed by the ACF intervention but their notification is actually tied to their home district and thus not counted in the notifications being used for the evaluation).

4) In the methods, it seems that access to healthcare is being determined by authors. It is unclear how they make this decision since the defnitions are based on features of the health system and population that are not always reported. For instance, the average distance that patients travel to a health facility is not a standard feature reported in many “study setting” descriptions. If there are assumptions being made, these need to be stated.

5) Inclusion and inclusion criteria used in various phases of the search are not clear. What were the inclusion criteria for the abstract/title review versus the main text review? Which step of the PRISMA flow diagram reflects the title/abstract versus the full text review? By convention, I would guess that the first box that simply says “articles not eligible for inclusion” reflects the title/abstract review, but neither the methods nor the flow diagram explain what was being selected for at this phase. Similarly, the inclusion criteria for the second phase of the search are not clear. Line 224-225 “qualitative or quantitative data on the impact of ACF on TB health seeking behaviour” is vague. Since two papers were excluded for not reporting KAP results in a specific way, the authors presumably had specific criteria for what they included here.

6) The narration of the included studies/articles appears inconsistent between the figure and different parts of the results.

- The PRISMA diagram shows 13 articles reporting routine TB case notification rate analyses and 1 article reporting behavioral outcomes.

- Table 1 shows 12 articles reporting TB case notification rates and 1 article reporting KAP outcomes.

- The first paragraph of the results say 12 articles reported case notification data, 1 from the initial search reported behavioral outcomes, and then from the second phase of the search, “three studies were eligible for inclusion as KAP/qualitative manuscripts.”

- The results section “KAP and qualitative studies” starting at line 379 says “Of the four publications identified, two were excluded from further analysis.”

Please make all of the reporting consistent. There seems to be a discrepancy about whether there were 12 or 13 articles identified in the first phase of the search and 2, 3, or 4 identified during the second phase of the search. Moreover, since one expects the first paragraph of the results to summarize the identification of studies, it would make sense to report the total number of included studies identified by the second search in this paragraph rather than only the number “eligible” (presumably this means after the abstract review?) and then wait until much later in the results to report the number actually included.

7) Since the two searches are part of a single systematic review, then I believe that the additional qualitative/behavioral identified through the second phase of the search should included in Table 1. Ideally the second phase of the search and what it yielded would also be part of the flow diagram, although I realize this would make the diagram complex (although there is a lot of blank space right now, so if the boxes were more efficiently positioned, the additional search process might fit).

8) In Figure 3, the horizontal axis has to be labeled. It is not clear what periods reflect pre-implementation, implementation, and post-implementation phases.

9) There needs to be some consistency in how different periods are referred to. Pre-post studies typically include a period prior to implementation of an ACF intervention, and a period during which an ACF intervention period is being implemented; evaluating a period after the intervention has stopped being implemented is also possible, but not as common. The paper uses various terms to refer to different periods, and I am not always sure what is being referred to. In line 333, does “post-intervention” refer to the period during the intervention or after the intervention has been completed? Similarly, in lines 432 and 438, does “following community ACF” (lines 432, 438) mean during the implementation of a community ACF program, or after the program has ended? Figure 4 uses the terms “baseline” and “endline” while figure 5 uses “before” and “After.”

10) Based on the data availability statement, all data are available in the manuscript and appendices. However, the quantitative data extracted from the individual articles does not appear anywhere in the manuscript. There is no way to tell the sizes of these studies or the numbers of cases detected. There is no way to “back into” these numbers from the CNR ratios reported.

11) I think that paragraphs 2 and 3 of the discussion are critical because they help the reader to interpret what the CNRs reported in the results mean. However, I find these paragraphs difficult to understand. What is the impact that one expects to see if ACF has a positive impact on health-seeking behavior? What impact might strengthening of diagnostic services have? There are a lot of references back to specific tables and figures, which is awkward in the discussion. Rather than referring back to specific tables and figure in the results, I think that these paragraphs should try to articulate the big picture of what one expects to see and then offer an interpretation of what was observed.

12) The discussion would benefit from a discussion of how the authors believe that studies should evaluate indirect impacts on behavior. I am not convinced that routine CNR ratios are the most meaningful measure of this, but I would be open to an argument to convince me. How would one ideally measure impact of an ACF program on care-seeking behavior, stigma, or norms? Pre-post testing? Longitudinal testing on the same individuals (i.e. the same individuals are evaluated before and after the program is implemented in their community) or cross-sectional testing in on population samples before and after implementation? What kinds of instruments should be used? What is the role of qualitative or mixed-methods research?

Minor comment

In reviewing prior research that has established the direct impact of ACF, it is worth mentioning the Cochrane review that encompassed this topic in addition to the more recent review by Burke et al and the older review by Kranzer et al (Mhimbira et al. Interventions to increase tuberculosis case detection at primary healthcare or community-level services, 2017).

6. PLOS authors have the option to publish the peer review history of their article (what does this mean?). If published, this will include your full peer review and any attached files.

**Do you want your identity to be public for this peer review?** For information about this choice, including consent withdrawal, please see our Privacy Policy.

Reviewer #1: **Yes: **Olivia Biermann

Reviewer #2: No

---

## [Decision Letter · Decision Letter 1]

18 Oct 2021

PGPH-D-21-00221R1

Do community-based active case-finding interventions have indirect impacts on wider TB case detection and determinants of subsequent TB testing behaviour? A systematic review

Dear Dr. Feasey,

Thank you for submitting your manuscript to PLOS Global Public Health. After careful consideration, we feel that it has merit but does not fully meet PLOS Global Public Health’s publication criteria as it currently stands. Therefore, we invite you to submit a revised version of the manuscript that addresses the points raised during the review process.

We look forward to receiving your revised manuscript.

Kind regards,

Stefan Kohler

Academic Editor

Journal Requirements:

Additional Editor Comments (if provided):

Reviewers' comments:

Reviewer's Responses to Questions

**Comments to the Author**

1. If the authors have adequately addressed your comments raised in a previous round of review and you feel that this manuscript is now acceptable for publication, you may indicate that here to bypass the “Comments to the Author” section, enter your conflict of interest statement in the “Confidential to Editor” section, and submit your "Accept" recommendation.

Reviewer #2: (No Response)

2. Does this manuscript meet PLOS Global Public Health’s publication criteria? Is the manuscript technically sound, and do the data support the conclusions? The manuscript must describe methodologically and ethically rigorous research with conclusions that are appropriately drawn based on the data presented.

Reviewer #2: Yes

3. Has the statistical analysis been performed appropriately and rigorously?

Reviewer #2: Yes

4. Have the authors made all data underlying the findings in their manuscript fully available (please refer to the Data Availability Statement at the start of the manuscript PDF file)?

Reviewer #2: Yes

5. Is the manuscript presented in an intelligible fashion and written in standard English?

Reviewer #2: Yes

6. Review Comments to the Author

Reviewer #2: Please see attachment - the character limit for the comment box seems to be set too low.

7. PLOS authors have the option to publish the peer review history of their article (what does this mean?). If published, this will include your full peer review and any attached files.

**Do you want your identity to be public for this peer review?** For information about this choice, including consent withdrawal, please see our Privacy Policy.

Reviewer #2: No

---

## [Decision Letter · Decision Letter 2]

15 Nov 2021

Do community-based active case-finding interventions have indirect impacts on wider TB case detection and determinants of subsequent TB testing behaviour? A systematic review

PGPH-D-21-00221R2

Dear Dr. Feasey,

We're pleased to inform you that your manuscript has been judged scientifically suitable for publication and will be formally accepted for publication once it meets all outstanding technical requirements.

Within one week, you'll receive an e-mail detailing the required amendments. When these have been addressed, you'll receive a formal acceptance letter and your manuscript will be scheduled for publication.

An invoice for payment will follow shortly after the formal acceptance. To ensure an efficient process, please log into Editorial Manager at https://www.editorialmanager.com/pgph/ click the 'Update My Information' link at the top of the page, and double check that your user information is up-to-date. If you have any billing related questions, please contact our Author Billing department directly at authorbilling@plos.org.

Kind regards,

Stefan Kohler

Academic Editor

Additional Editor Comments (optional):

Reviewers' comments:

Reviewer's Responses to Questions

**Comments to the Author**

1. If the authors have adequately addressed your comments raised in a previous round of review and you feel that this manuscript is now acceptable for publication, you may indicate that here to bypass the “Comments to the Author” section, enter your conflict of interest statement in the “Confidential to Editor” section, and submit your "Accept" recommendation.

Reviewer #2: All comments have been addressed

2. Does this manuscript meet PLOS Global Public Health’s publication criteria? Is the manuscript technically sound, and do the data support the conclusions? The manuscript must describe methodologically and ethically rigorous research with conclusions that are appropriately drawn based on the data presented.

Reviewer #2: Yes

3. Has the statistical analysis been performed appropriately and rigorously?

Reviewer #2: Yes

4. Have the authors made all data underlying the findings in their manuscript fully available (please refer to the Data Availability Statement at the start of the manuscript PDF file)?

Reviewer #2: Yes

5. Is the manuscript presented in an intelligible fashion and written in standard English?

Reviewer #2: Yes

6. Review Comments to the Author

Reviewer #2: (No Response)

7. PLOS authors have the option to publish the peer review history of their article (what does this mean?). If published, this will include your full peer review and any attached files.

**Do you want your identity to be public for this peer review?** For information about this choice, including consent withdrawal, please see our Privacy Policy.

Reviewer #2: No
